# Mobile App for Enhanced Anterior Cruciate Ligament (ACL) Assessment in Conscious Subjects: “Pivot-Shift Meter”

**DOI:** 10.3390/jpm14060651

**Published:** 2024-06-18

**Authors:** Edmundo Berumen-Nafarrate, Ivan Rene Ramos-Moctezuma, Luis Raúl Sigala-González, Fatima Norely Quintana-Trejo, Jesus Javier Tonche-Ramos, Nadia Karina Portillo-Ortiz, Carlos Eduardo Cañedo-Figueroa, Arturo Aguirre-Madrid

**Affiliations:** 1Star Medica Chihuahua Hospital, Perif. de la Juventud 6103, Fracc. El Saucito, Chihuahua 31110, Mexico; 2Faculty of Medicine and Biomedical Sciences, University Autonomous of Chihuahua (UACH), Chihuahua 31110, Mexico; iramos@uach.mx (I.R.R.-M.); lrsigala@uach.mx (L.R.S.-G.); fatimanorely@gmail.com (F.N.Q.-T.); javiertonche22@gmail.com (J.J.T.-R.); a309342@uach.mx (N.K.P.-O.); ccanedo@uach.mx (C.E.C.-F.); 3Department of Orthopedic Surgery, Star Medica Chihuahua Hospital, Perif. de la Juventud 6103, Fracc. El Saucito, Chihuahua 31110, Mexico; arturoaguirremadrid@yahoo.com.mx

**Keywords:** ACL, machine learning, mobile application, pivot-shift, rotational stability

## Abstract

Anterior cruciate ligament (ACL) instability poses a considerable challenge in traumatology and orthopedic medicine, demanding precise diagnostics for optimal treatment. The pivot-shift test, a pivotal assessment tool, relies on subjective interpretation, emphasizing the need for supplementary imaging. This study addresses this limitation by introducing a machine learning classification algorithm integrated into a mobile application, leveraging smartphones’ built-in inertial sensors for dynamic rotational stability assessment during knee examinations. Orthopedic specialists conducted knee evaluations on a cohort of 52 subjects, yielding valuable insights. Quantitative analyses, employing the Intraclass Correlation Coefficient (ICC), demonstrated robust agreement in both intraobserver and interobserver assessments. Specifically, ICC values of 0.94 reflected strong concordance in the timing between maneuvers, while signal amplitude exhibited consistency, with the ICC ranging from 0.71 to 0.66. The introduced machine learning algorithms proved effective, accurately classifying 90% of cases exhibiting joint hypermobility. These quantifiable results underscore the algorithm’s reliability in assessing knee stability. This study emphasizes the practicality and effectiveness of implementing machine learning algorithms within a mobile application, showcasing its potential as a valuable tool for categorizing signals captured by smartphone inertial sensors during the pivot-shift test.

## 1. Introduction

The knee is the largest and most complex joint in the human skeleton, essential for walking, supporting body weight, taking off, and receiving jumps, considered one of the joints most subjected to constant stress, for which it has a set of ligaments that give it its biomechanical skill: the anterior cruciate ligament (ACL), posterior cruciate ligament (PCL), medial collateral ligament (MCL), and lateral collateral ligament (LCL), as well as secondary or accessory stabilizers. The anatomy of the ACL is made up of two bundles: the anteromedial (AM) bundle mainly prevents anterior to posterior displacement, while the posterolateral (PL) bundle stretches in extension and prevents rotational instability of the knee. These bundles work synergistically during knee movement, presenting tension during extension and laxity during knee flexion, providing a role as the main stabilizer of the joint. Knee ligament injuries can occur due to a sudden axial load on the knee, combined with valgus stress and rotational force around the tibia. An injury to the ligament complex results in a loss of stability; therefore, an early clinical diagnosis of instability is imperative [1,2,3,4,5]. 

To assess the integrity of this structure, imaging techniques and even arthroscopy itself are used. Clinical evaluation includes tests such as the Lachman test, anterior drawer test, and pivot-shift (PS) test, considered the “gold standard” for assessing knee rotational function after an injury. There are also scales based on questionnaires from the International Knee Documentation Committee (IKDC) and the Tegner Activity Scale (TAS). However, these are subjective measurements [4,5].

On the other hand, to quantify knee laxity, various systems, such as the KT1000 and KT2000 arthrometers (MED metric Corp., San Diego, CA, USA), have been developed to measure sagittal laxity. However, methods for evaluating rotational laxity involve a range of techniques, including ambulatory devices, instrumented boots, magnetic resonance imaging (MRI), electromagnetic sensors, robotic technology, and navigation systems. While these technological advancements are impressive, it is crucial to consider their practicality, affordability, and comfort when implementing them in a clinical setting. The use of technologies to quantify the pivot maneuver is currently being studied; infrared markers are one of them, as well as the use of sensors like accelerometers to detect the signal generated by the pivot maneuver, with the disadvantage of high cost of the hardware [6,7,8,9,10,11,12,13,14,15,16,17].

In a study by Tanaka et al. [18] comparing the diagnostic reliability of the quantitative evaluation of the pivot-shift test, using an electromagnetic measurement system for anterior cruciate ligament deficiency against an accelerometer system (KiRA), an application of Image analysis using external devices, no significant differences were found between these measurements; both measurement methods were able to detect ACL laxity; however, it is important to highlight the need to use additional devices, generating extra costs for the patient [19].

Among the available tests, the PS test stands as the sole clinical evaluation that effectively correlates with knee joint hypermobility. The PS test subjectively assesses pivot change and has been the focus of efforts to quantify bone movements and correlate them with laxity grading. However, it is important to acknowledge that the unrestricted and dynamic nature of the PS maneuver introduces significant kinematic variability, and a need for careful consideration during the evaluation process.

The use of built-in sensors in smartphones for diagnostic applications has been increasing in recent years [20,21,22,23,24], as well as the development of machine learning algorithms to solve problems regarding pattern recognition [25,26,27,28,29]. To address the limitations of subjective evaluations, Pivot-Shift Meter (PSM) has been developed: a novel mobile application capable of recording signals acquired by the integrated inertial sensors of the mobile device during the pivot-shift maneuver. This study aims to process the information gathered through the PS maneuver by the development of filtering algorithms and machine learning, to determine the reproducibility of the pivot-shift maneuver between testers based on the acquired signals. Furthermore, this study further aims to establish an ACL instability/laxity grade by utilizing the KT-1000 arthrometer as the gold standard tool.

## 2. Objectives

Evaluate the efficacy of AI for analyzing and categorizing signals obtained from inertial sensors during pivot-shift tests to identify the most effective approach for integration into mobile software.Evaluate the reliability and effectiveness of the developed software in assessing knee joint stability through intraobserver and interobserver analyses.Compare the results obtained from the developed application with those from the KT-1000 arthrometer, a gold standard tool for assessing knee laxity.Investigate the potential of the developed software to enhance the precision and objectivity of clinical evaluations of ACL injuries and other joint conditions.

## 3. Materials and Methods

### 3.1. Population

The study sample size for the trial was obtained from 66 volunteer students who participated in a control event held at the School of Medicine and Biomedical Sciences of the Universidad Autónoma de Chihuahua. Delving into the demographics, considering a total population of 156 4th-year medicine alumni, the study sample size was selected with a confidence level of 95% and a margin of error of 10%.

Inclusivity was paramount in our participant selection process, encompassing individuals aged between 18 and 37 to construct a well-rounded representation of both young and middle-aged adults. Striving for gender balance, our cohort comprised 34 women and 32 men, acknowledging potential gender-related nuances in the study’s outcomes. Before enrollment, all participants underwent a thorough informed consent process, affirming their comprehension of the study’s objectives, methodologies, and associated risks. As the study focused on healthy subjects, individuals with pre-existing orthopedic conditions, injuries, or any other known health issues were excluded from participation. The study protocol had been approved by the ethics committee of “Christus Muguerza del Parque” hospital.

The evaluation phase was orchestrated by a proficient team of six evaluators, boasting a diverse range of expertise and experience in orthopedic surgery. Comprising two orthopedic surgeons with over 25 years of seasoned practice, complemented by a practitioner with 3 years of experience, and further enhanced by the contribution of three budding orthopedic surgeons with less than 3 years of practice. This diverse panel was entrusted with conducting a meticulous examination, which included the rigorous administration of the pivot-shift (PS) test, capturing and documenting pertinent data.

Among the initial 66 participants, only the data from 52 participants were deemed usable for analysis due to technical errors in data saving or mislabeling of the samples. The excluded data were not included in the subsequent analysis to ensure the accuracy and reliability of the results.

### 3.2. Measurements

The development of the Pivot-Shift Meter (PSM) application stemmed from a dual focus: firstly, to capture and record the signal generated by the gyroscope in the mobile device; and secondly, to seamlessly store and organize this information in an online database. The initial version of the application was developed using the program MIT App Inventor 2.

The application successfully achieved two key goals:Signal capture: The PSM application recorded angular velocity (rad/s) data corresponding to the three axes of movement on the mobile device at a sampling rate of 100 Hz, resulting in 500 data points captured in 5 s—this high-frequency data acquisition aimed to capture precise and detailed measurements of the pivot-shift maneuver. X-axis data was used because it corresponds best to the movement of the leg when executing the maneuver concerning the position of the cell phone (Figure 1A).Data storage: The recorded data was securely saved in a database, which was configured to provide a user-friendly experience for the physicians involved in the study. This database facilitated the physician’s ability to review and follow up on each case (Figure 2). During the study, the pivot-shift maneuver was performed three consecutive times on each test subject. The mobile device was placed on the anteromedial aspect of the tibia, approximately two fingers away from the tibial tuberosity. To ensure secure placement, the mobile device was connected to a sports-type cellular armband with the PSM application pre-installed, allowing for accurate recording of the maneuver.

### 3.3. Data Recording in the PSM Application by the Evaluator

Patient Data Recording: patient initials, age, gender, height, and weight (Figure 2).Placement of the Cell Phone: the cell phone with the PSM application installed is placed on the tibial tuberosity of the patient, securing it approximately two fingers below the patella with an elastic band, ensuring that the device is slightly tilted towards the medial aspect of the tibia (Figure 1B).Execution of the Pivot Maneuver: the evaluator holds the ankle on the medial side with the hand corresponding to the patient’s leg, and with the opposite hand, the posterior part of the leg is held at the level of the tibial head, rotating slightly medially. Subsequently, the leg is flexed until reaching a 90° angle.Results Recording: the evaluators are instructed to perform two maneuvers and save the results obtained with the PSM application. Additionally, the application allows for adding observations, clinical classification according to IKDC criteria, results of digital arthrometry (KT-1000), results of imaging studies, arthroscopic images, and notes with relevant clinical information (Table 1).

**Table 1 jpm-14-00651-t001:** Input data acquired in the study.

**Input Data**	**Description**
Patient Demographics	Initials, age, gender, height, weight
Gyroscope Signals	Angular velocity (rad/s) recorded at 100 Hz (500 data points over 5 s)
Clinical Classification	Results of the IKDC criteria, digital arthrometry (KT-1000), and other relevant clinical information.

In addition to the PSM data, the KT-1000 arthrometer was used to measure the anteroposterior displacement of the knee joint in millimeters. This measurement provided an assessment of the laxity in the anteromedial (AM) bundle of the ACL. Measurements were performed on both legs of each participant, and the results were compared to determine the degree of injury. The development of the laxity grade table was inspired by the need to provide a comprehensive assessment tool for the resulting data. The objective measurements were obtained through the KT-1000 arthrometer, which indicates the millimeter displacement of the knee joint, and a grading system, “Del Parque” Classification (CDP), which categorizes laxity into grades 1, 2, and 3. In recognizing the importance of identifying cases without significant knee laxity, an additional grade was added to the classification table, grade 0. This grade signifies a stable knee with minimal or no laxity.

With this scale, a knee laxity grade is obtained by measuring both legs of a patient and determining the difference between these results (Table 2). The table ensures consistency in grading PSM data by providing clear criteria for each grade.

### 3.4. Statistical Analysis

The Intraclass Correlation Coefficient (ICC) was used to compare repeatability among the evaluators of the data acquired by the PSM when performing a pivot-shift test. MatLab (version 2023) was the software used for this purpose.

The ICC is used to find correlations within a single class of data rather than two different classes of data. Is a value that goes from 0 to 1 and acts as a measure of the reliability of the ratings for data that exist in groups of quantitative variables.

From each X signal of each evaluator, features of the signal that agree with the amplitude in radians per second (rad/s) and location in time of each positive peak in centiseconds (cs) were obtained, as seen in marks PI-III in Figure 1C and summarized in Table 3.

Characteristic vectors (CV) were formed by data provided by each peak of each signal:

Subject 45: [137.54, 179.86, 156.83]; [111.00, 267.00, 448.00]

Once this set of samples was arranged, two types of comparisons were made: an intraobserver and an interobserver.

### 3.5. Classification Algorithm

The main objective was to find which features of the signal could help us to determine a degree of laxity. For this analysis, 35 of the 52 samples were used to maintain similar groups in the number of samples of each grade, composed of maneuvers classified with the KT-1000 in the different grades, from 0 to 3 according to the CDP grading system (Table 2). To test the classification algorithms (CA), a test group of 9 cases were separated, 3 for each degree of injury, except for grade 3 cases, since in our entire sample there were only 2 occurrences of this grade of injury, thus leaving 23 samples as our training group.

Different methods were tested, including support vector machine (SVM), decision tree, random forest, and logistic regression, using first raw signals and then pre-processed signals.

The first step of the signal processing was to normalize every signal to values between 0 and 1 to be further analyzed only by its morphological features.

Next, the area indicated as the “pivot” in Figure 1C was extracted from each sample in its normalized values (NVs) and its original values (OVs). Features considered for the CA were obtained from the NVs and the OVs of the “pivot” region of the signal as follows: the standard deviation (SD) from the NV, the maximum value (Max), the range plus average of the signal values (R + $\bar{X}$), and the sum of squares divided by the amount of data in the signal ($\frac{\sum x^{2}}{N}$) from the OV. These values were taken to test the AI.

## 4. Results

### 4.1. Signal Interpretation

From the samples obtained, it was identified that the X-axis corresponds best to the movement of the leg when executing the maneuver with regard to the position of the cell phone. Thus, it was possible to relate portions of the graph to leg movements. Three sectors were identified and related: flexion, extension, and “pivot” (Figure 1C).

### 4.2. ICC Analysis

The intraobserver ICC analysis showed a global average of 0.95 and 0.71 in terms of the time between maneuvers and the maximum amplitude of each one, respectively. On the other hand, the interobserver analysis yielded 0.94 and 0.66, allusive to the time and maximum amplitude between each maneuver, respectively. From these correlation results, the samples were grouped into classes determined by the speed of the maneuver reached, and then each sample was analyzed to determine the degree of laxity. Initially, 28 participants were classified according to the potential degree of laxity, which was distributed as in Table 4.

### 4.3. Classification Algorithm

The raw signals corresponding only to the X-axis were classified with AI; the test accuracy is reported in Table 5. The highest result, 40%, is above the chance level (25%) in three of the four methods.

Searching for better results, the signals were processed to isolate the “Pivot” segment following the process shown in Figure 3A–F. Then, the processed signals were used to create CAs based on AI, first by classifying each maneuver in a class, then classifying each signal grade of laxity according to their class; results of performance metrics can be seen in Table 6. The performance metrics were obtained with the test group using the machine learning methods to classify the three degrees of laxity proposed by the CDP grading system; these are summarized in Table 7. Confusion matrices were generated out of these results and are reported in Figure 4.

### 4.4. Highlights

Through comprehensive intraobserver and interobserver analyses, this study aimed to evaluate the reliability and effectiveness of the developed software. The results revealed strong agreement among evaluators, underscoring the software’s reliability in assessing knee joint stability. Furthermore, a comparative analysis was conducted between the results obtained from the developed application and those from the KT-1000 arthrometer, a gold standard tool in the field. This comparison highlighted the potential clinical utility of the software in providing quantitative assessments of knee laxity.

Overall, the study demonstrates the successful development of a mobile application for assessing knee joint stability. The observed agreement among evaluators and the promising comparison with the KT-1000 arthrometer underscore the potential of the software to enhance the precision and objectivity of clinical evaluations for ACL injuries.

## 5. Discussion

Technological advancements have played a critical role in the development of new medical techniques and treatments, particularly in the field of orthopedics, allowing for less invasive procedures and more effective clinical analysis methods during joint physical examinations. While the pivot-shift test is considered a cornerstone in assessing knee laxity, its clinical application comes with several significant limitations. Performing the test and evaluating the results are subjective tasks prone to errors. This subjectivity is a relevant factor contributing to uncertainty among observers and intraobservers, limiting the reliability of the obtained data. Additionally, the development of tools and the use of technology in clinical settings to optimize diagnosis, prognosis, and medical treatment have revolutionized the concept of healthcare. Inertial sensors like accelerometers and gyroscopes in smartphones offer a novel, convenient, accessible, and comparatively affordable option for diagnosing and quantitatively evaluating ACL injuries in knees. The KT-1000 arthrometer cannot be compared with the PS maneuver due to the nature of the biomechanics of each maneuver. Nevertheless, in the present study, the Pivot-Shift Meter app was able to accurately classify acquired signals, providing quantitative dynamic diagnostic information of rotational instability that the gold standard arthrometer cannot offer.

According to the Landis and Kotch criteria, the time in which the maneuvers were performed reached an almost perfect value (>0.8) both in the intraobserver analysis and in the interobserver analysis, which may be because the evaluators adapted test after test to achieve three maneuvers in a row within five seconds [30]. Meanwhile, the substantial correlation obtained between the intraobserver analyses, regarding the amplitude in which the peaks were detected, may be because each evaluator has a unique way of executing the maneuver; anatomical factors may influence both the evaluator and the examination subjects, such as height and physical condition. This suggests that a stricter standardization of the pivot maneuver is necessary, ideally given by the same application. As Nicola Lopomo et al. mention in their systematic review, a generalized pivot change maneuver is still desirable, which was demonstrated with the ICC analysis of the maximum amplitude of each maneuver [30,31,32,33,34]. Also, Hoshino et al. describe how a standardized technique improves measurement accuracy; thus, once this standardization is applied, it is possible to obtain a higher correlation [31].

None of the AI methods were successful in the classification of the raw signal. Thus, preprocessing the signal to obtain only the “pivot” segment was necessary, which can be explained as the hysteresis of the inertial sensor after the knee joint reduction; hence, the higher the SD of the data in this portion of the signal, the higher the movement detected in the knee joint reduction. This approach proved to be successful, as three out of four of the AI methods were able to classify the signals correctly according to the performance metrics. Moreover, the performance metrics suggested that two different AI methods were successful for each classification algorithm, SVM for the “Pivot” portion classification and decision tree for the laxity grading classification.

Regarding the comparison against the KT-1000, a relationship of 27/38 classifiable cases was determined, equaling 71.05%. This ratio was low since eight cases could not be correctly determined by PSM, belonging to classes 0 and 1. Therefore, there remains a ratio of 27/30 (≈90%) cases. These three maneuvers were part of our training group, improving our AI generalization capabilities. 

Compared to other ways to quantify pivot-shift tests, the approach proposed in the present work has some advantages, such as the fact that it is very simple to use, as the only other requirement besides the device itself is a proper way to fix the phone in position; as mentioned before, a simple armband was used for this purpose. Also, this software has no trouble being installed in both commercial operative systems [6,32,33,34,35].

Some other authors have approached pivot-shift quantification by the use of inertial sensors, especially accelerometers. Vaidya et al. [9] also conducted a study with mobile devices using these sensors. Even when they had satisfactory results, it was noted that these sensors tend to add more noise to the signal; this is because what they measure (acceleration) depends on more variables of a higher order compared to the speed measured by gyroscopes [8,36,37,38,39].

Most authors that approach the quantification of the pivot test through accelerometers seem to agree on a method to interpret a signal acquired by the pivot test with only one morphological characteristic in the graph, which allows them to detect subluxation and reduction of the joint, equivalent to the “pivot” segment of the signals acquired in this study. Nevertheless, this method could be enhanced by the use of machine learning algorithms [40,41,42,43,44,45]. Labbe et al., and more recently Yañez-Diaz et al., implemented SVM algorithms to signals acquired with accelerometers in order to grade the PS phenomenon objectively, reporting acceptable results, which can be compared to the results of the algorithm to organize the signals by class. Notwithstanding, the SVM was not the best method to sort the signals by grade [44,45].

Results of initial studies showed a low correlation between intraobserver analysis regarding the amplitude of the pivot-shift effect graphed in the application due to the variables presented by each evaluator during the maneuver. This led to the initiation of another current multicenter and international case study aiming to analyze, classify, and follow more than 400 tests performed by specialist physicians from five different countries, including Bolivia, Chile, Colombia, Ecuador, and Mexico, using the mobile application PSM on patients with ACL injuries.

## 6. Limitations

While the PSM demonstrated promise, acknowledging certain limitations is crucial for a comprehensive interpretation of the findings. The study’s reliance on a cohort of 52 healthy subjects may raise questions about the general applicability of the results to broader populations. The subjective nature of the pivot-shift maneuver, despite attempts at standardization, highlights the ongoing challenge of achieving consistent execution among evaluators.

Furthermore, the comparison with the KT-1000 arthrometer revealed limitations in the PSM’s ability to classify certain cases accurately. The 71.05% relationship in classifiable cases indicates room for improvement, particularly in accurately determining cases belonging to classes 0 and 1. These discrepancies emphasize the need for further refinement and standardization of the PSM application. Despite these limitations, this study underscores the advantages of the PSM approach, highlighting its simplicity, accessibility, and cost-effectiveness. The ongoing multicenter international case study involving specialists from various countries aims to address the limitations identified in the initial study as well as the implementation of more sophisticated methods of classification, such as the “Dual-branch collaborative learning network” or the “Generate Adversarial-Driven Cross-Aware Network” used by Zhang W. et al. [46,47]. The observed variability in intraobserver analysis, particularly about the amplitude of the pivot-shift effect, underscores the need for thorough investigation and software adaptation to accommodate diverse users. 

Our purpose is to optimize the software’s adaptability for each user, and the motivation behind conducting this study lies in the importance of diagnosis and postoperative monitoring of ACL injuries due to the potential repercussions, such as early joint degeneration and knee instability. Therefore, digital arthrometry appears to be the path toward developing precise and numerical evaluation methods for rotational instability, with all the advantages that early diagnosis and classification can provide before any surgical treatment.

## 7. Conclusions

The development of this innovative software marks a significant stride forward for orthopedists, ushering in a future where only a mobile device is needed, eliminating additional expenses for examiners and patients. This advancement not only introduces new technologies but also enhances the scope of comprehensive care, facilitating timelier diagnoses and more effective treatments tailored to each case. By integrating injury degree classification with specific surgical recommendations for each grade, we aim to furnish clinicians with a practical tool to discern the most appropriate approach for individual patients. Furthermore, the successful integration of machine learning methods as CAs into a mobile application for assessing and classifying signals from inertial sensors within a cellphone during a pivot-shift test underscores the potential of technology in clinical evaluations. A decision tree turned out to be the best machine learning method to implement as a CA with the data analyzed in the present work.

In general, the application of technology such as inertial sensors in smartphones and neural networks can serve as a valuable tool to enhance the precision and objectivity of clinical evaluations of ACL injuries and other joint conditions. Continued research and refinement of these tools are imperative to ensure their accuracy and utility in clinical practice.

It is crucial to acknowledge that no measurement system is flawless, and there are inherent limitations in their clinical application. One such limitation is the necessity of adhering to standardized pivot maneuver techniques, as variability in technique may impact the accuracy of the measurements obtained.

Further studies are warranted to validate the measurement accuracy of the pivot, using the application across different patient populations and clinical scenarios. This will ascertain the application’s ability to identify ACL injuries at various stages of the injury and evaluate the utility of inertial sensors in clinical practice.

## Figures and Tables

**Figure 1 jpm-14-00651-f001:**
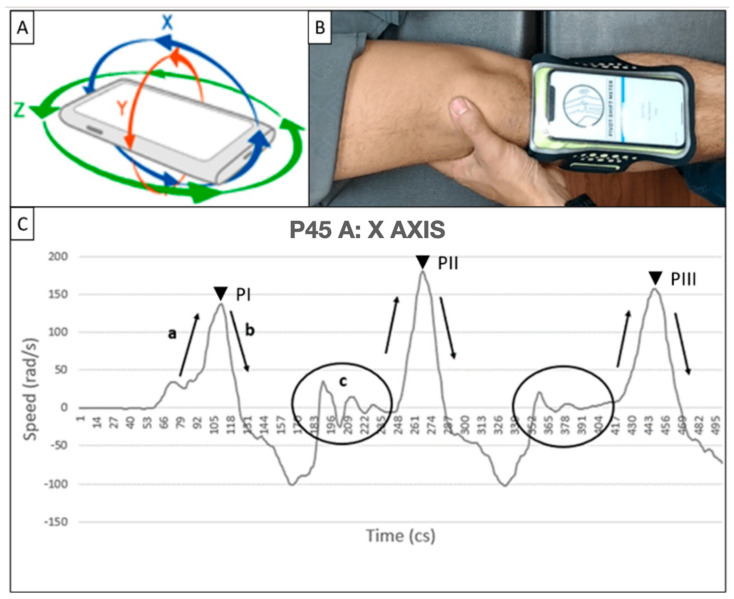
(**A**) X, Y, and Z axis of movement of gyroscopes integrated in a mobile device. (**B**) Placement of the mobile device on the anteromedial face of the tibia, two finger widths from the tibial tuberosity and attached with a sports-type cell phone bracelet. (**C**) X-axis signal can highlight the sequence of the pivot-shift maneuver: (a) flexion movement, (b) extension movement, and (c) joint reduction or “pivot”. It is also noticeable that this pattern repeats, giving a clue to the start of the next maneuver due to the peaks (PI–III).

**Figure 2 jpm-14-00651-f002:**
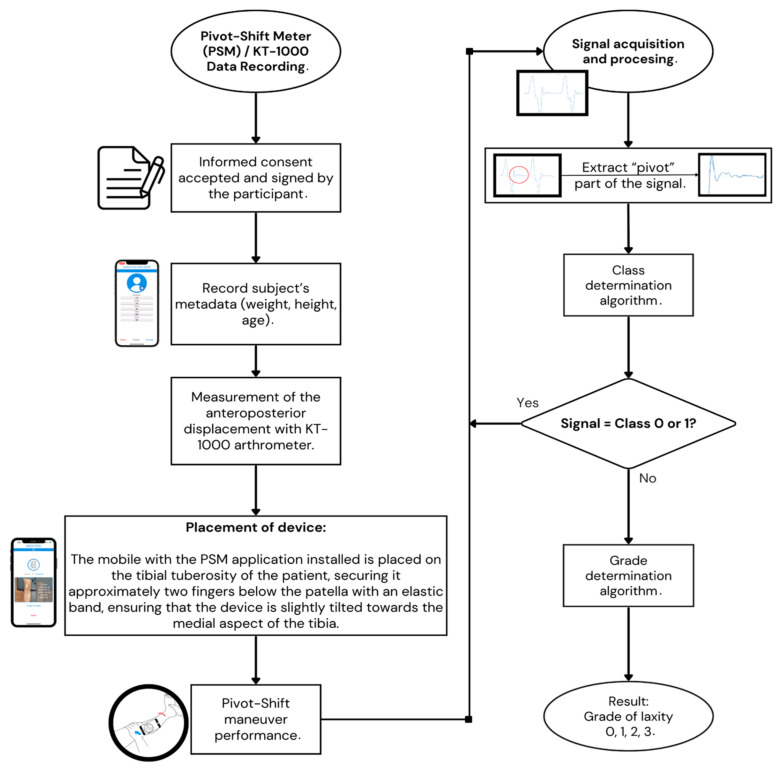
Data Recording PSM App and signal processing flowchart.

**Figure 3 jpm-14-00651-f003:**
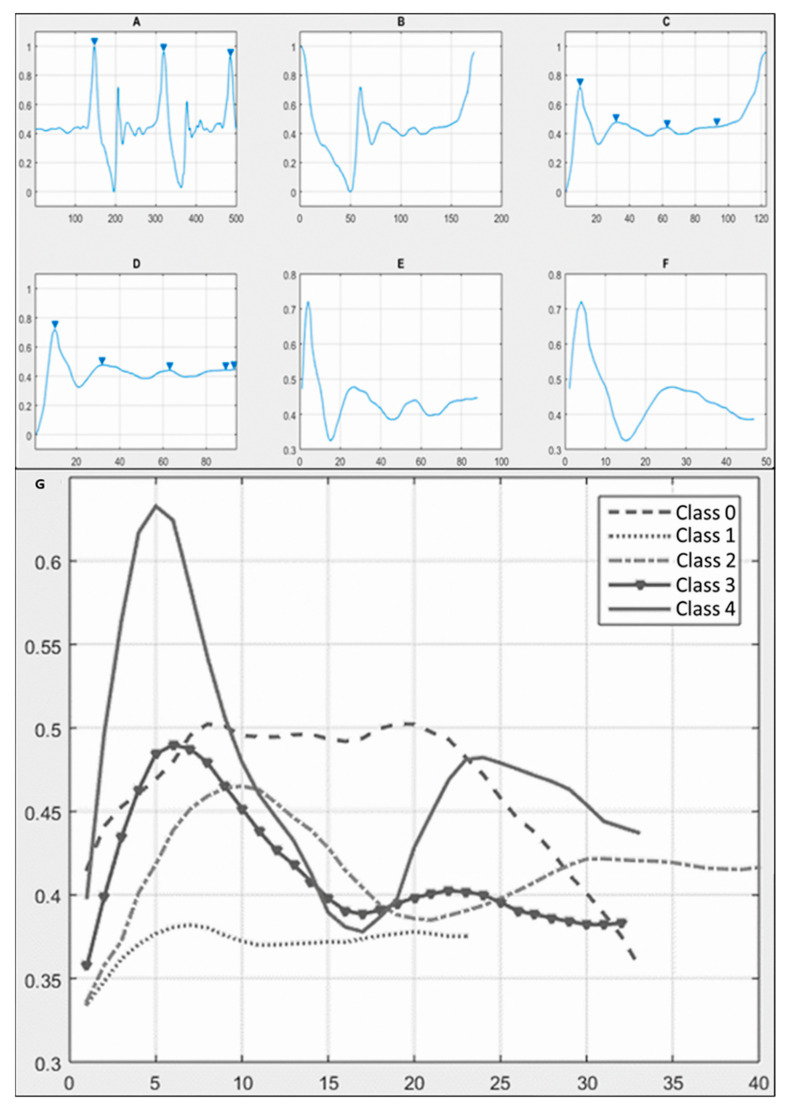
(**A**–**F**) Signal processing. After normalizing the signal, it was segmented to obtain the important portion of data, which corresponds to the “pivot” (**F**). (**G**) Graphic difference between classes: 1, 2, 3, and 4 have a sinusoidal-based waveform, while class 0 has a plateau shape.

**Figure 4 jpm-14-00651-f004:**
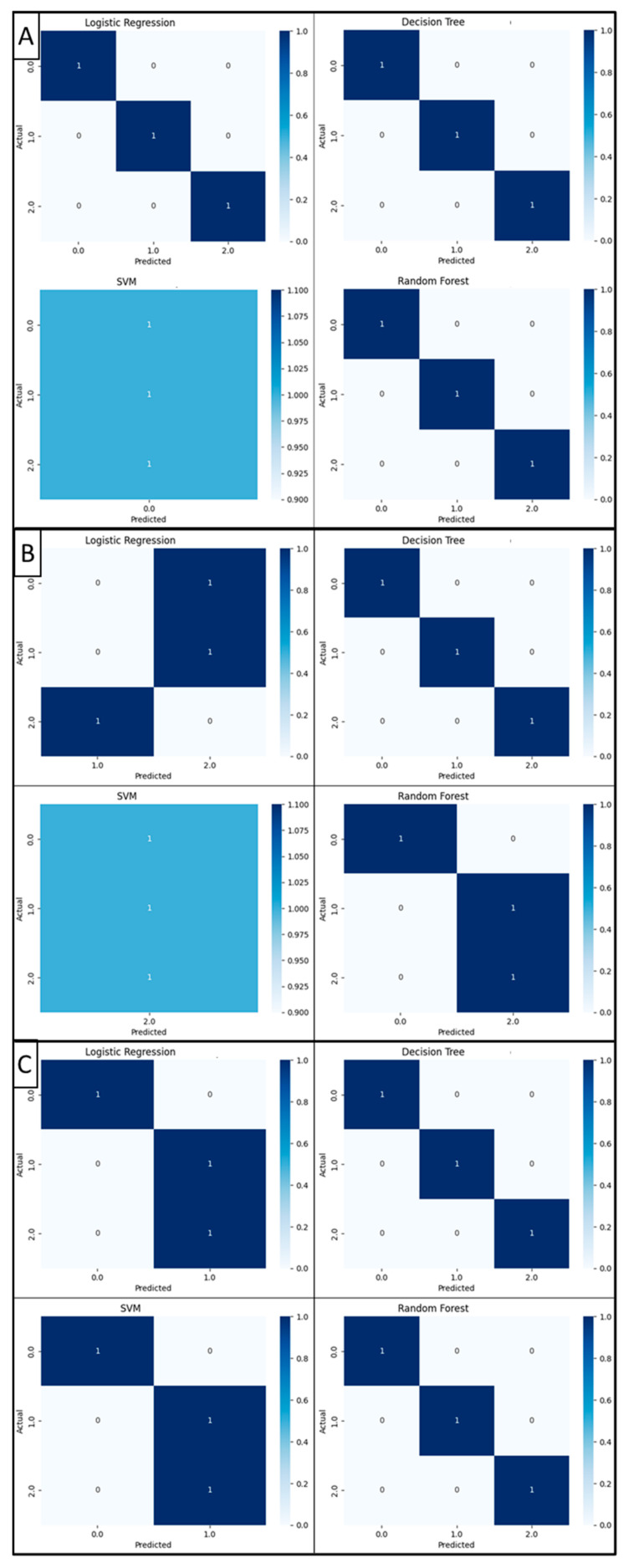
Confusion matrices of AI tested for class 2 (**A**), class 3 (**B**), and class 4 (**C**).

**Table 2 jpm-14-00651-t002:** CDP: grading system for knee laxity based on difference in millimeters measured by KT-1000 [15].

Grade	Difference in mm	Meaning
0	0–2	Almost null laxity
1	3	Low laxity
2	4–5	Considerable laxity
3	>6	High laxity

**Table 3 jpm-14-00651-t003:** Summary of signal features arrangement for the ICC analysis.

	Evaluator A			Evaluator N
	Peak I	Peak II	Peak III	Peaks I, II, III
Subject n − 1	…			…
Subject 45	Time of detection (cs)			…
	111.00	267.00	448.00	
	Amplitude (rad/s)			
	137.54	179.86	156.83	
Subject n + 1	…			…

**Table 4 jpm-14-00651-t004:** Classification of participants according to their potential degree of laxity.

Grade of Laxity	Number of Subjects
0	10
1	9
2	9
3	2

**Table 5 jpm-14-00651-t005:** Accuracy of ML used to classify by grade of laxity with raw signals.

Machine Learning Method	Accuracy (%)
Logistic Regression	40
Support Vector Machine	20
Decision Tree	40
Random Forest	40

**Table 6 jpm-14-00651-t006:** Performance metrics of every ML method used to classify “Pivot” signals by class.

Machine Learning Method	Accuracy (%)	Recall (%)	Precision (%)	F1-Score
Logistic Regression	28.57	28.57	28.57	0.2857
Support Vector Machine	100	100	100	1.0
Decision Tree	85.71	85.71	85.71	0.8571
Random Forest	85.71	85.71	85.71	0.8571

**Table 7 jpm-14-00651-t007:** Performance metrics of ML methods used to classify by grade of laxity per class.

Class 2	Accuracy (%)	Recall (%)	Precision (%)	F1-Score
Logistic Regression	100	100	100	100
Support Vector Machine	33	33	33	33
Decision Tree	100	100	100	100
Random Forest	100	100	100	100
**Class 3**	**Accuracy (%)**	**Recall (%)**	**Precision (%)**	**F1-Score**
Logistic Regression	0	0	0	0
Support Vector Machine	33	33	33	33
Decision Tree	100	100	100	100
Random Forest	66	66	66	66
**Class 4**	**Accuracy (%)**	**Recall (%)**	**Precision (%)**	**F1-Score**
Logistic Regression	66	66	66	66
Support Vector Machine	66	66	66	66
Decision Tree	100	100	100	100
Random Forest	100	100	100	100

## Data Availability

Derived data supporting the findings of this study are available from the corresponding author on request: soporte@ormeds.com.mx.

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
