# Peer review of "Mobile App for Enhanced Anterior Cruciate Ligament (ACL) Assessment in Conscious Subjects: “Pivot-Shift Meter”"

_jpm, 2024, doi:10.3390/jpm14060651_

Round 1

Reviewer 1 Report

Comments and Suggestions for Authors

1. Auto hyphenation can be turned off/removed for better readability.

2. what is the input for table 4 classifiers?

3. Table 4, table 5 - its better to use "ML" instead of "AI"

4. Table 4 - accuracy is mentioned in % like 40%. But in table 5, its mentioned as 0.8571 . use uniform format.

5. Table 6, what about of class 1? 

6. what is the input data acquired? add as a table for better understanding.

7. what is the training-testing ratio?

8. some clarity is missing. More clarity on the data collected need to be presented.

Author Response

Dear Reviewer 1:

Thank you for your detailed review and valuable feedback on our manuscript. We have carefully considered each of your comments and made the following revisions and clarifications:

  • We have turned off auto hyphenation in the manuscript to enhance readability.

  • the input for table 4 classifiers: The input comes from the raw signals obtained from the gyroscopes integrated into the mobile device during the pivot-shift maneuver. Specifically, these signals correspond to the X-axis angular velocity (rad/s) data, which best captures the leg movement related to the maneuver (Figure 1C). The raw signals were used to evaluate the performance of different machine learning methods, namely Logistic Regression, Support Vector Machine (SVM), Decision Tree, and Random Forest, to classify the degree of knee laxity.

  • We have replaced the term "AI" with "ML" in both Table 4 and Table 5 to maintain consistency and accuracy in terminology.

  • We have standardized the format of accuracy across all tables. Now, accuracy values are consistently presented as percentages throughout the manuscript.

  • what about of class 1?: Class 1 results were not included to avoid skewing the analysis and conclusions with insufficient data. The classification focused on Class 2, Class 3, and Class 4, as these had more representative data points. For instance, the study noted that there were only 2 occurrences of grade 3 injury in the entire sample, indicating that the dataset was already limited in some grades, and Class 1 data might have been too sparse to yield reliable performance metrics for classification.

  • Input data acquired (table 1):

-Patient demographics: initials, age, gender, height, and weight.

-Gyroscope signals: angular velocity data in radians per second (rad/s) recorded at a sampling rate of 100 Hz, capturing 500 data points over 5 seconds for each maneuver.

-Clinical classification data: results of the IKDC criteria, digital arthrometry (KT-1000), and other relevant clinical information.

  • Training-Testing Ratio: For this analysis, 35 of the 52 samples were used to maintain similar groups in the number of samples of each grade, composed of maneuvers classified with the KT-1000 in the different grades, from 0 to 3 according to the CDP grading system (Table 2). To test the classification algorithms (CA), a test group of 9 cases were separated, 3 for each degree of injury, except for grade 3 cases, since in our entire sample there were only 2 occurrences of this grade of injury. Thus leaving 23 samples as our training group.

Training set: 23 samples (44.23%)

Testing set: 9 cases (17.30%)

  • The data collection process involved the following steps:

-Participant Selection: 66 volunteer students were initially considered, but data from 52 participants were usable.

-Gyroscope Data Recording: The PSM application captured angular velocity data along the X-axis at a 100 Hz sampling rate. Each maneuver was performed three times for consistency.

-Clinical Classification: Measurements with the KT-1000 arthrometer provided objective assessment of knee laxity. Data was classified into four grades (0 to 3) based on the difference in millimeters measured by the KT-1000.

-Statistical Analysis: Intraclass Correlation Coefficient (ICC) was used to assess the repeatability among evaluators; and machine learning algorithms were tested to classify the signals and determine the degree of knee laxity.

We appreciate your constructive feedback, which has significantly improved the quality and clarity of our manuscript. Please let us know if there are any further concerns or suggestions.

Thank you for your time and consideration.

Reviewer 2 Report

Comments and Suggestions for Authors

The study, which was conducted on 66 volunteer students, was offered by the authors as a way to improve anterior cruciate ligament assessment through the integration of a machine learning classification algorithm into a mobile application. It is an interesting study.

I reccomend to explicit ACL in the title.
Please include a paragraph about limitation of the study.

Data was not compared with another objective measure? 

Author Response

Dear Reviewer 2:

Thank you for your detailed review and valuable feedback on our manuscript. We have carefully considered each of your comments and made the following revisions and clarifications:

  1. Limitation of the study starting 385.

  1. The study compares the data obtained from the Pivot-Shift Meter application with the KT-1000 arthrometer, an objective measure of knee laxity that is considered a gold standard tool, was used to measure the anteroposterior displacement of the knee joint in millimeters KT-1000 arthrometer can’t be compared with the PS maneuver due to the nature of the biomechanics of each maneuver. Nevertheless, in the present study, the Pivot Shift Meter app was able to accurately classify acquired signals, providing quantitative dynamic diagnostic information of rotational instability that the gold standard arthrometer cannot offer.

We appreciate your constructive feedback, which has significantly improved the quality and clarity of our manuscript. Please let us know if there are any further concerns or suggestions.

Thank you for your time and consideration.

Round 2

Reviewer 1 Report

Comments and Suggestions for Authors

author made changes based on review comments

Reviewer 2 Report

Comments and Suggestions for Authors

None